# Audiogenic Seizures and Social Deficits: No Aggravation Found in Krushinsky–Molodkina Rats

**DOI:** 10.3390/biomedicines11092566

**Published:** 2023-09-18

**Authors:** Anastasiya Rebik, Nadezda Broshevitskaya, Syldys Kuzhuget, Pavel Aleksandrov, Kenul Abbasova, Maria Zaichenko, Inna Midzyanovskaya

**Affiliations:** 1Institute of Higher Nervous Activity and Neurophysiology, Russian Academy of Sciences, 117485 Moscow, Russia; rebik-a-a@ihna.ru (A.R.); mariya-zajchenko@yandex.ru (M.Z.); 2Faculty of Biology, Lomonosov Moscow State University, 119234 Moscow, Russiaakenul@gmail.com (K.A.)

**Keywords:** epilepsy, seizures, autism spectrum disorder, autistic disorder, anxiety, phenotype, animal models, motivation, KM rats, Wistar rats

## Abstract

Epilepsy or epileptic syndromes affect more than 70 million people, often comorbid with autism spectrum disorders (ASD). Seizures are concerned as a factor for social regression in ASD. A stepwise experimental approach to this problem requires an animal model to provoke seizures and monitor subsequent behavior. We used rats of the Krushinsky–Molodkina (KM) strain as a validated inbred genetic model for human temporal lobe epilepsy, with recently described social deficiency and hypolocomotion. Generalized tonic-clonic seizures in KM rats are sound-triggered, thus being controlled events in drug-naïve animals. We studied whether seizure experience would aggravate contact deficits in these animals. Locomotor and contact parameters were registered in “the elevated plus maze”, “socially enriched open field”, and “social novelty/social preference tests” before and after sound-provoked seizures. The triple seizure provocations minimally affected the contact behavior. The lack of social drive in KM rats was not accompanied by a submissive phenotype, as tested in “the tube dominance test”, but featured with a poor contact repertoire. Here, we confirmed our previous findings on social deficits in KM rats. The contact deficiency was dissociated from hypolocomotion and anxiety and did not correlate with seizure experience. It was established that experience of rare, generalized tonic-clonic convulsions did not lead to an impending regress in contact motivation, as seen in an animal model of genetic epilepsy and comorbid social deficiency. One of the oldest animal models for epilepsy has a translational potential to study mechanisms of social behavioral deficits in future neurophysiological and pharmacological research.

## 1. Introduction

Epilepsy and epileptic syndromes are among the main neurological comorbidities of autism spectrum disorder (ASD). They are diagnosed in 7% of autistic children and in as many as 19% of autistic adults [1]. Seizure history is reported as a risk factor for intellectual disabilities and social regresses [2,3,4]. Experience of seizure is thought to provoke hidden epileptogenesis, rewiring the brain and lowering the threshold for the next convulsions.

Since there is a significant rate of the ASD/epilepsy comorbidity in human patients [5,6], ASD-like phenotypes should also be more expectable in epilepsy-prone laboratory animals, if compared to the entity of seizure-resistant ones. So, to find an animal model for ASD, it is reasonable to screen animal epilepsy models for ASD-like phenotypes. A possibility to assess putative epilepsy-related effects on social deficits would bring an advantage of accumulated epileptology knowledge to a relatively new field of neurology of ASD.

Indeed, there are several animal models of epilepsy with reported social deficits [6,7,8,9,10]. On the other hand, the connections between ASD and epilepsy are neither unidirectional nor simple, both in clinical practice and basic research. In a number of animal epilepsy models [11,12,13,14], minor or no deficits in conspecific communications were registered. In a mouse model [15], ASD-like and epilepsy-like behaviors did not demonstrate any additivity, but rather counteracted. In clinics, prescriptions of antiepileptic drugs do not help to mitigate ASD phenotypes, but might even worsen the behavioral adaptation of patients [16]. So, ASD-like behavioral phenotypes might not be directly caused by seizure experience, but may partly share common neuropathological mechanisms. For a stepwise experimental approach to study the ASD/epilepsy interaction, one needs to provoke seizure fits according to a controlled schedule. Animal models are indispensable for such experimentations.

Recently, we reported social contact deficits in rats of Krushinsky–Molodkina (KM) strain [17]. This Wistar-derived inbred [18] strain has been used as a model for human temporal lobe epilepsy and vascular pathology from the late 1940s [19] to currently [20,21,22,23]. All KM rats are susceptible to audiogenic seizures (AGS) of the maximal score, in response to sound provocations. The seizures fits start from a paroxysmal running and quickly proceed to the clonic and tonic-clonic seizures, followed by profound postictal catalepsy. Importantly, generalized seizures can be triggered in drug-naïve animals. It leaves out the question of possible focal effects, as well as side effects of chemical convulsants, and allows for an unbiased investigation of post-convulsive behavior.

Seizure-naïve KM rats demonstrated social contact deficits, as we reported previously [17]. Namely, KM rats developed a highly exaggerated freezing response to a presence of social stimuli, spent less time in a close proximity to unfamiliar conspecifics and demonstrated a lower number of contacts, as compared to control Wistar rats. It is not known whether a start of seizure would further aggravate social deficits in these epileptic rats. We enrolled a battery of sociability tests, to study the very first stage of ASD/epilepsy interaction: behavioral consequences of rare, generalized tonic-clonic seizure in drug-naive KM rats. The test paradigms varied in spatial design and also enrolled different rats as the social stimuli, to level out possible influences of individual traits of “guest” animals, as well as of spatial context. In the rat, as well as in other species, social interactions of conspecifics are highly dependent on gender, feasibility of aggressive actions, sensory modalities available for interaction, dominant ranks of interacting animals, presence of a shelter, and general available space. Although we did not aim to study gender effects here, the other factors listed above were varied in our experimental routine. Both spatial conditions available for interaction of conspecifics and sufficient time of experiments were considered as important factors. The battery of behavioral tests used also allowed us to compare the obtained results with literature data, since there is no single and commonly accepted set-up for sociability tests in laboratory animals yet [24].

The relatively mild scheme of seizure fits provocation, used in the present experiments, let us keep focused on the beginning of the active epileptogenesis and assess its effect on social contact motivation. The schedule of rare seizures was chosen to escape kindling, which would recruit new neuronal loops and hyper-activate limbic structures [25]. Limbic activation would in turn trigger its own behavioral consequences [26]. As it was shown recently, a fourteen-day schedule of AGS kindling in KM rats led to evident neurodegenerative processes, such as an irreversible hippocampal loss of granular and hilar mossy cells, as well as a reduced proliferation of inferior colliculus neurons [27]. In a shorter scheme, seven-day AGS provocation design, the cell losses were rescued by autophagy [27]. An overspread of epileptiform activity in the magnocellular neurons of the hypothalamus, induced by audiogenic kindling, evoked neuroendocrine changes in the hypothalamic-neurohypophysial system [28]. So, frequent and numerous seizure fits provocations produce a complex response, which needs to be dissected and analyzed separately in further experimentations.

## 2. Materials and Methods

There is an increasing attention to experimental studies of sociability in laboratory animals. No single experimental design is commonly accepted, but a variety of non-standardized tests are used [24,29]. Social interaction might be assessed in a couple of freely moving animals, placed in a simple or enriched context [12,29,30,31]. Alternatively, “guest” animals might be fixed with tether and collar assemblies [30,32], separated in a compartment of the test arena [33,34,35], or encaged [36,37,38]. The social stimuli might be isolated by partitions, which specifically limit or restrict visual, audial, haptic, or olfactory sensory inputs involved in animal interactions [39,40,41]. Assessment of behavioral parameters in freely moving pairs of animals represented the most difficult case for automated tracking, but the obstacle seems to be resolved soon with algorithms of machine learning [29]. Currently, social interaction of two freely moving animals in laboratory conditions implies a high amount of tracking artifacts. A disadvantage of this design is a possibility of aggressive actions for the experimental animals. More frequently, experimenters let the test animal choose locomotor strategies freely, whereas the social stimuli are localized in a compartment or encagement. This limits behavioral repertoire of social stimuli, but not the test rodents. The locomotion of test animals might be tracked automatically, with a possibility to select virtual zones of interest. Usually, conventional trackers allow an automated detection of freezing behavior as well. Nowadays, automated tracking is usually complemented with offline semi-automated detection of contact as well as investigatory and comfort behavioral elements, performed by blinded experts. For more details on social tests in laboratory rodents, a reader is referred to the recent reviews [24,29,42].

We chose a battery of behavioral tests for sociability (“the socially enriched open field”, “three chambered social preference/social novelty tests”, “two choice social preference/social novelty tests”, see Section 2.3 and Section 2.4), to detect intricacies of conspecific contacts in a robust way. The use of multiple tests minimized possible artifacts related to spatial context [29], as well as a putative bias related to individuality of social stimuli. The experimental schemes described below (Section 2.3 and Section 2.4) assumed the free movement of experimental animals so that they could adjust the intensity of social approaches to encaged unfamiliar conspecifics.

### 2.1. Animals

The experiments were performed in 43 KM and 44 Wistar male rats in total. The animals were 5 months old at the start of experimentation and weighed 300–350 g. The animals were housed 3–5 in a cage, with standard pellet food and water ad lib. Behavioral tests were scheduled with a 6–10 days interval between the manipulations, unless otherwise stated.

The main behavioral tests were performed in two rounds, pre- and post-provocation of AGS.

After the 1st set of the tests (i.e., the “elevated plus maze”, “socially enriched open field”, “social preference/social novelty”), sound provocations were scheduled for half of the experimental animals. For this purpose, the rats were randomly assigned to the “sound provocation subgroup” (*n* = 8 for both strains) and the “sham provocation” subgroup (*n* = 8 for both strains). The sound provocation/sham was performed in 3 sessions, separated by 6–8 days. The sound administration induced audiogenic seizure fits in all KM rats, but not in any of the Wistar rats, as described below (Section 2.6). In a week after the last sound stress, the main behavioral tests (i.e., the “elevated plus maze”, “socially enriched open field”, “social preference/social novelty”) were repeated.

Additional tests (“two choice social preference/social novelty”, “social dominance tube test”, “novel object recognition test”) were run in the animal cohorts, not subjected to the sound stress.

### 2.2. Elevated Plus Maze

A standard “elevated plus maze” test (EPM) was performed, as described in [43]. In short, the apparatus consisted of 2 open arms (10 cm × 46 cm), 2 closed arms (10 cm × 46 cm × 16 cm), and a central plate (10 cm × 10 cm). The maze was raised 90 cm from the floor surface. Video camera was installed about 120 cm above the maze. The experiment was recorded using a computer in the next room.

The rats (*n* = 16 for each strain) were placed individually on a central plate, pointing its head to the open arm and turning their backs to the experimenter; landing served as a signal to start tracking. The animals were allowed to explore the maze for 10 min. The video archives were analyzed off-line, as described below (see Section 2.7). The time spent and distance travelled in open and closed arms, rearing, short (facial) and long (whole body) grooming, stereotyping (if any), defecation and urination (if any) were counted.

### 2.3. Sociability Tests

The rat cohorts were tested in different spatial modifications of the social preference tests, to compare the results with literature data. The main idea of the social tests was to allow a freely moving test animal to explore the available arena with an encaged unfamiliar conspecific. As it was shown previously, normal rats demonstrate a clear preference to stay in a neighborhood of sheltered newcomer rats, rather than nearby empty cages [24,34,35].

#### 2.3.1. The Two-Choice Test for Social Preference

The experiment was conducted in a T-shaped maze with a closed starting section. The main part of the maze (sized as 75 × 14 × 32 cm) was divided into 6 zones. The left and right side compartments for stimulus rats (sized as 12.5 × 14 cm) were separated from the main part of the maze by a double grid. During testing, the maze was illuminated with diffused light (40 W) to reduce the stress of the animal (modified from [33]).

The test consisted of two sessions, each lasting 10 min. Before the experiment, the test rat was placed in an empty maze for 2 min adaptation.

In the 1st session (social preference), a stimulus rat, unfamiliar to the test animal, was placed in one of the side compartments of the T-shaped maze, separated by a double lattice, while the second side compartment was left empty (Figure 1(b.1)). During the 2nd session (social novelty), another stimulus animal (“new stimulus”) was placed in the previously empty side compartment, while the “old stimulus” remained on its place (Figure 1(b.2)). Between the sessions, the test animal was placed in a carrier. After each test animal, the floor of the maze was thoroughly cleaned with a 10% ethanol solution. Video registration was performed and analyzed off-line. Times spent in a neighborhood of each compartment, the contact time (defined as the time of a snout contact with the stimulus box), and freezing response were registered. The contacts were further defined as the short (<6 s in duration) and the prolonged ones (>6 s in duration), as suggested by S. Wagner and colleagues [34]. The experimental groups consisted of 7 KM males and 8 Wistar rats. The test was not repeated in further experimentation.

#### 2.3.2. Socially Enriched Open Field Test

The test was modified from Castro et al. [35]. The apparatus consisted of the black plastic rectangular arena (120 cm × 120 cm), surrounded by 40 cm walls and lighted. A video camera was attached at about 2 m above the floor. Four dark plastic boxes, routinely used for rats’ transportation in the laboratory (sized as 18 cm × 26 cm × 16 cm as W × L × H, with evenly perforated walls and lids, and with two holders as 1.5 cm × 8 cm holes), were placed at the middle of each wall (Figure 1b).

One of the 4 boxes contained an unfamiliar male Wistar rat (of the same age and weight), used as a social stimulus. The other 3 boxes contained a portion of bedding from the stimulus rats’ home cage, to unify the olfactory cues. The location for the stimulus box was changed for every new experimental session, to counterbalance possible environmental cues unrelated to the experiments.

After the 20–30 min of habituation, the test animals were individually allowed to investigate the arena for 20 min; their trajectories were monitored online from an adjacent room and digitized for off-line analysis. Time spent in a neighborhood of each box, the contact time (defined as the time of a snout contact with the stimulus box), the total track lengths, and local track lengths in a neighborhood of each box, were registered. The contacts were also defined as the short and the prolonged ones [34].

The experimental groups consisted of 16 KM males and 16 Wistar rats. After the triple sound/sham provocation, the test was repeated in all groups.

#### 2.3.3. Three-Chambered Social Preference/Social Novelty Test

The 3-chambered test for social preferences/social novelty was used as described earlier by other researchers [24,44] and us [17]. The experiments were preceded by 20–30 min of habituation, when the test animals were allowed to explore the complete set-up, excluding the social stimuli. The apparatus consisted of a black arena (60 × 60 cm, with black wooden walls 60 cm high) and partitions dividing the box into two chambers and a start rectangular compartment between them (Figure 1(a.1)). Each partition had an opening providing free passages between the sections. Two dark plastic perforated containers, the same as described above (see Section 2.3.2) were placed in each chamber. Firstly, the installation included an empty box and a box that contained an unfamiliar male Wistar rat of the same age and weight, used as a social stimulus. The test animals were allowed to explore the environment for 10 min.

For the 2nd session (“social novelty test”) the empty box was replaced with an identical container with a new rat to test the reaction to social novelty (Figure 1(a.2)). The test rats were intact between the sessions. The video registration lasted another 10 min. The behavior was recorded as described above. Below, these classes are called tests for “social preferences” and “social novelty”.

The sessions were video-recorded with a digital camera linked to a computer in the next room; the recordings were digitized and analyzed off-line automatically.

The results of the 1st trial of the “social preference/social novelty” in a smaller cohort have been recently published [17].

The tests were repeated after the sound/sham provocations (see Section 2.6) in all the groups (in total, in 13 KM and 16 Wistar males).

### 2.4. Social Dominance Test

The social dominance tube test (TSD) was carried out using a 150 cm-long transparent plexiglass tube with an internal diameter of 6 cm [45]. The experiment was carried out for three days: on the first two days, the animals were allowed to explore a new environment and pass through the pipe in both directions, on the third day, the testing itself was carried out. Prior to the experiment, the rats were weighed and assigned to pairs so that in each of them were unfamiliar Wistar and KM male rats of a similar body weight. The rats were placed at the opposite ends of the pipe (Figure 1(d.1)) and allowed to move (Figure 1(d.2)). The “winner” was considered the animal that was able to push the opponent back and exit from the opposite end of the tube (Figure 1(d.3)). Subsequently, the video recording of the experiment was analyzed. We recorded the latency to wins, the number of pushes, forward propulsions, and retreats [46].

The experimental groups consisted of 20 KM males and 16 Wistar rats, not subjected to the AGS provocations.

### 2.5. Two-Object Novel Object Recognition

The Novel Object Recognition [47] task was conducted in an open field arena (60 × 60 × 60 cm, the same apparatus as described above at Section 2.3.3, but without the partitions), supplemented with two different kinds of new objects (plastic toy bricks sized as 5 × 5 × 8 cm). Both objects were similar in height and volume, but different in shape and appearance. During 15 min of habituation, the animals were allowed to explore the empty arena. One hour after habituation, the animals were exposed to the familiar arena with two identical objects placed at an equal distance. Next, the test rat was allowed to explore the open field in a presence of the familiar object and a novel object to test recognition memory. Each of the sessions lasted 10 min, instead of the standard 3 min, to account for the hypolocomotion of KM rats. In addition to the standard indicators described above, the following were also registered: contacts with the object—sniffing the object and active contact with the object—touching and moving the object with front paws, turning the object over, and transferring in the teeth.

The experimental groups consisted of 12 KM males and 12 Wistar rats, not subjected to the AGS provocations.

### 2.6. AGS Provocation

After the 1st set of behavioral tests, a sound provocation scheme [48,49,50,51] was randomly assigned to a half of KM and Wistar cohorts. Briefly, individual rats were placed in a sound-attenuated box (47 × 47 × 47 cm) and exposed to a loud (80–120 dB) sound (electric bell ringing) for 90 s, or until the tonic phase of the seizure was reached (score 9 according to the scale proposed by P. Jobe [52]). The sound evoked initial freezing and then an orientation response in Wistar rats. All KM individuals developed the fully blown AGS (Figure 2) with a latency <15 s. The sham-provocations were the placement into the same box without a sound exposure.

The triple AGS provocations or sham-provocations were scheduled in a week interval, to escape kindling of the seizures (see Introduction). In a week post-provocations, the “elevated plus maze” test (Section 2.2) and the main sociability tests (Section 2.3.2 and Section 2.3.3) were repeated.

### 2.7. Automated Tracking

All sessions were recorded on video using a digital camera connected to a computer in the next room. The recordings were digitized and analyzed automatically off-line (track length, freezing, time spent in virtually defined marked areas) and off-line by blind experts. Registered parameters were the contact time, rearings, and short/long grooming episodes.

The free ToxTrac 2.98 software was used for off-line tracking of moving animals in all tests. The xld identification algorithm (Ssi.Rep.) was used. The algorithm used allowed us to track the rostral part of the body of a freely moving animal (which was important when evaluating social contacts). The following measurements were made automatically: the time spent in certain zones, the total distance travelled in certain zones, and the number and time of freezing episodes [53].

### 2.8. Statistical Analysis

The normality was checked by the Kolmogorov–Smirnov criterion (Basic Statistics, Descriptive Statistics section). The main method for assessing the significance of differences between the experimental and control groups was the Mann–Whitney U-test, and within-group effects were estimated by the Wilcoxon test. Bonferroni corrections were applied for between-group comparisons, under assumption of two independent behavioral domains (locomotor behavior and contact behavior). The differences were considered statistically significant at *p* ≤ 0.05. The main calculation of statistics took place using the Statistics 10.0 program, graphical images were created using Python 3.7.3 using the SciPy library.

## 3. Results

### 3.1. Elevated Plus Maze

The 1st EPM (*n* = 16 for KM rats, *n* = 16 for Wistar rats) test resulted in a similar between-strain difference as reported earlier by us [17] and others [54]. It is suggestive for a reduced exploratory drive and increased anxiety in KM rats. Namely, the total track length was shorter, as well as the open arms time; the number of rearings was lower and the number of facial grooming was higher, with a domination of freezing behavior in KM rats (see Appendix A).

The repeated, post-provocation EPM revealed that the horizontal activity of KM rats was increased as compared to the first testing (Appendix A) and thus reached the level of control of the Wistar rats (Figure 3; Appendix A). Vertical activity was still deficient (Appendix A), and facial grooming enhanced (Appendix A) in KM as compared to Wistar rats. The control rats did not change their locomotion between the EPM tests (Appendix A).

The sound provocation per se did not exert a gross effect: horizontal and vertical locomotion did not differ between the subgroup of seizure-experienced and sham-provoked rats (Figure 3a,b; Appendix A). The AGS-experienced KM rats did not differ significantly from their sham-provoked strain mates (Appendix A).

### 3.2. Different Paradigms of “Social Preference/Social Novelty” Tests

#### 3.2.1. Two Choice Social Preference Test

The social deficits in KM males were consistently seen in the different modifications of sociability tests, with different unfamiliar stimulus rats proposed for contacting. The “two-choice test” was the simplest setup. In this paradigm, the rats mainly demonstrated short (<6 s) contacts, presumably reflecting general curiosity but not a social drive [34]. The prolonged, developed (>6 s) social investigations were almost absent in this condition (Appendix A). Short contacts summed up to a lower time in KM rats (Figure 4a; Appendix A), as compared to Wistar ones (Appendix A). The spontaneous behavior in the test was highly aberrant with profound freezing response (Figure 4b,e; Appendix A) and thus did not provide sufficient information on locomotion or investigatory drive.

#### 3.2.2. The “Socially Enriched Open Field Test”

The “socially enriched open field” set-up provided more space for the freely moving animals and a possibility to investigate the arena, the container with an encaged unfamiliar conspecific, or three empty containers.

The pre-provocation tests were also marked with an increased number of freezing episodes and general hypolocomotion in KM rats (Appendix A). The social drive deficits can be seen as a decreased percentage of contact trajectories (see Appendix A), paralleled by a lack of well-developed (>6 s) contacts (Appendix A).

The post-provocation tests demonstrated persistent social deficits in KM rats: the reduced numbers of the short (Figure 5f) and prolonged (Figure 5g) contacts, lower percentage of the contact trajectory (Appendix A) and lower percentage of the time spent in a stimulus neighborhood (Appendix A). No major difference was seen between the sound-stressed and non-stressed subgroups of the same strains (Figure 5i–m; see also Section 3.3, Appendix A).

#### 3.2.3. The Social Preference/Social Novelty Test

The pre-provocation tests demonstrated the very similar results as reported previously in a smaller cohort [17]. Briefly, KM rats escaped the chamber with an encaged conspecific during the “social preference” session, developing a low number of snout-to-box contacts (Appendix A), accompanied by pronounced freezing (Appendix A). The addition of a new conspecific (the “social novelty” session) emphasized the freezing response in KM but not Wistar rats (Appendix A) and provoked hypolocomotion paralleled by a low number of any social contacts (all *p’s* < 0.001, Appendix A).

The repeated (post-provocation) “social preference/social novelty” tests revealed normalized locomotion in KM rats as observed in the “social preference” session (Figure 6a), with no difference in a number of short (Figure 6b) or long (Figure 6c) contacts (but a decreased time spent in the stimulus chamber, Appendix A).

Again, the well-pronounced contact deficits in KM rats were clearly seen in the “social novelty” session. The addition of the second unfamiliar encaged conspecific led to a decreased contact activity in KM rats: the numbers of prolonged contacts with both “familiar” and “unfamiliar” stimuli reduced significantly (Figure 6e,f, respectively; see Appendix A).

The AGS experience demonstrated its mild activating effect in the “social novelty” session (see also below, Section 3.3): the subgroup with the triple provoked seizure fits tended to a higher number of developed and short snout-to-box contacts (both *p* = 0.06, Figure 7f; Appendix A) towards the “unfamiliar” guest rats. Nevertheless, even with this unexpected tendency to activation, the between-strain differences evidenced the contact deficits in the KM groups (Figure 6; Appendix A).

### 3.3. Effect of Seizures’ Experience

The sound provocation did not evoke AGS in control rats or exert a gross behavioral effect in this cohort.

Surprisingly, no major difference between the “latent AGS” and «experienced AGS» subgroups of KM rats was seen in the post-provocation social tests. Existing social deficits in KM cohorts were not aggravated by the triple provocation of AGS. Instead, a tendency to activation was seen. Namely, a moderate decrease in anxiety levels was seen as the lowered number of freezing episodes (Appendix A) and decreased facial grooming (Appendix A). Unexpectedly, the convulsive experience led to a tendency to enhance the contact behavior (see Section 3.2). Locomotion was not increased by the seizure experience. Instead, we observed a facilitation of locomotion by the repeated testing in KM but not Wistar rats (see below).

### 3.4. Effect of Repetitive Testing

The main behavioral tests (“elevated plus maze”, “socially enriched open field”, “three-chambered social novelty/social preference” tests) were performed twice, before and after the sound exposures (referred as the first and the second trials, or “pre-provocation”/“post-provocation”). For the control rats, we did not see a major difference between the first and second tests in the behavioral parameters registered (Appendix A). To overcome potential artifacts caused by the excessive freezing response and hypolocomotion, we used the percentages of time spent near the stimulus’ box and the percentages of path travelled around it (Appendix A).

As for the KM cohorts, the repeated testing helped to overcome the initial hypolocomotion (Figure 5a, Figure 6a, and Figure 7a; Appendix A), so that the two cohorts of epileptic rats almost reached the locomotor activity level of controls (Appendix A). Namely, no strain/subgroup difference was seen for the total distance travelled in the following tests: “elevated plus maze”, “three-chambered social preference test”, “socially enriched open field”.

The deficits in social contacts generally remained, seen in the decreased numbers of prolonged contacts with the stimulus (“socially enriched open field”, see Section 3.2.2; “social novelty” test, all *p* < 0.01, see Section 3.2.3).

### 3.5. Two Object Novel Object Recognition Test

Novel object recognition tests are used to assess investigatory drive and memory in laboratory rodents. We used the same apparatus as in the previous experiments (see Section 3.2.3), but without the partition, to check interaction with inanimate objects, in the same environmental context as the social interactions were observed. KM (*n* = 12) and Wistar (*n* = 16) demonstrated only a minor difference in the test: KM more often interacted with the new object (Figure 8(a.2,b.2)). The number of active contacts with the new object (i.e., touching with paws, transferring in teeth, or gnawing the objects) was higher in KM rats (Appendix A). The results also showed that the rats of the KM strain were not frightened by the novelty of small inanimate objects. They kept interacting with the object, while the control Wistar rats lost their interest quite soon.

### 3.6. Social Dominance “Tube” Test

Contact motivation deficits might be a result of a social defeat experience or a lack of social dominance. To test this, we run the test on a dominant behavior, so called “social dominance in a tube”. According to the results, KM rats won the test more frequently (Appendix A). It should indicate a greater level of dominance in the KM, as compared to the Wistar rats. At the same time, the latency to win was significantly longer in KM rats (Figure 9a; Appendix A). A more detailed behavioral analysis showed that the KM rats made a fewer number of pushes during the sparring and fewer retreat trials to escape (Figure 9b,d). KM prevented the opponents from moving forward through the pipe. At the same time, the number of forward propulsions was similar for the both groups (Figure 9c). Thus, Wistar and KM groups demonstrated different behavioral strategies in TSD. Wistar rats preferred an active strategy, as evidenced by the low latency to win and a larger number of pushes and retreat trials. On the contrary, KM rats chose a rather passive strategy characterized by slow movements, a lack of retreats, and gradual progression forward by the time of retreat trials of Wistars.

Thus, consistent social contact deficits were observed in KM rats, subjected to a battery of behavioral test. The active phase of epileptogenesis was started by a triple provocation of audiogenic seizures, and the presumed behavioral changes were monitored post-provocation. As a result of seizure provocations, as well as of multiple experimental sessions, we observed a mild behavioral activation in KM rats. Namely, hypolocomotor traits in KM rats were rescued up to the control level, together with a decrease in anxious behavior. Contact deficits were consistently seen both before the AGS provocation as well as post-provocation. Control groups of Wistar rats did not majorly change their behavior during the course of experimentations.

## 4. Discussion

Epilepsy-related social and intellectual regress is a major concern for ASD patients [3]. To study this phenomenon experimentally, we focused on the start point of active epileptogenesis in an animal model of comorbid epilepsy and behavioral deficits in KM rats. We observed that hypolocomotion might be dissociated from the contact deficits in KM rats and that the social deficiency persisted during the multiple tests.

The hypolocomotor trait of KM rats can be normalized by multiple exposures to the test. Namely, KM rats displayed evident hypolocomotion in all the first (i.e., pre-provocation) tests (namely, EPM, “socially enriched open field”, “social preference/social novelty”, see Section 3.1 and Section 3.2., but not in all the post-convulsive tests. The feature contrasted with the attention deficit hyperactivity disorder (ADHD)-like hyperlocomotor phenotype of other animal models of epilepsy comorbid with social deficits: prenatal valproate loading [55] and fast/slow kindling rat strain [56]. At the same time, the observed hypolocomotion was in line with that reported for a maternal immune activation model of ASD [57]. Remarkably, profound neuroinflammation also plays a role in the pathogenesis of audiogenic epilepsy in KM rats, since a number of pro-inflammatory cytokines were affected by sound exposure in KM rats [22]. It is interesting to check whether hypolocomotion would be seen in other animal models of neuroinflammation-associated ASD.

The locomotor response in KM rats might be restored almost to the normal level by an extra-long habituation to the experimental conditions, as seen from the current results (in the second trials of the “elevated plus maze”, “three-chambered social preference test”). It is possible that longer repetitive social exposures would normalize contacts with conspecifics, in its turn. In our design, the same tests were separated by 6 weeks; it remains a question whether a quicker habituation would be possible for these rats. Anyway, the habituation to the experimental set-ups in KM rats was obviously longer than the regular 20–30 min sessions needed for normal rats. The feature is strongly suggestive for a disadaptation to environmental novelty in KM rats.

The experimental design allows discrimination between the effect of seizures/sound stress and the habituation effects, by the comparison of sham-provoked and sound-provoked groups. Expectably, control Wistar rats did not show any major effects from the sound stress administration (Appendix A). Surprisingly, there was only a minimal effect of the seizures’ experience in the KM rats. The severe but rare seizures exert rather a mild activation in KM rats, seen as decreased bouts of freezing and short grooming (Appendix A; Section 3.3), as well as a tendency toward contact activation (Figure 7f–h and Appendix A). No other behavioral consequences of the triple AGS provocation were seen in our experiments.

The tendency to AGS-related mild contact activation (Appendix A) might partly parallel a facilitating effect of the electroconvulsive therapy. The method is widely used in clinical practice to treat major depression, Parkinson’s disease, and/or schizophrenia [57,58,59,60]. The mechanisms are not fully understood, but at least partly it acts via an activation of brain aminergic metabolism [61,62,63]. Brain monoamines are imbalanced in KM rats [12,21,23], so one may hypothesize a facilitated turnover of brain amines as a putative mechanism for the observed behavioral activation. It would be irrational to recommend electroconvulsive therapies to treat ASD; however, there might be a mechanism to overcome the hypolocomotion and contact deficits.

The KM rats demonstrated a lack of social motivation in all the experimental setups used. The three-chambered “social novelty” test (Section 3.2.3, Figure 6g,h) appears to be the most sensitive one. We might hypothesize that this particular setup still leaves a possibility for spatial behavior but does not allow escaping the proximity of encaged “guest” rats. Also, it produced a high spatial density of conspecifics. It is known from clinical practice that ASD patients have particular difficulties in multiple social interactions at a time.

As it was shown previously, the contact behavior of freely moving rodents (rats or mice) did not depend on the locomotion of the encaged stimulus conspecific inside the container [34]. However, the contact behavior implies individuality in the interactions, putatively involving audial, tactile, and/or olfactory modalities. The contact behavior of test rats depended on a space available for interaction, as well as on ability to touch and smell the guest rats [39]. A close spatial confinement of interacting rats led to decreased contact motivation, while their partition by barred walls preserved it. Tactile interactions were essential for social reward, whereas visual contacting via a glass screen was not [39]. In our experiments, perforated walls allowed the olfactory inspections of all the containers, which were unified in smelling the bedding compound from the guest rats’ home cages. So, the only presumed difference between the containers was a presence of an encaged conspecific. The communication windows (1.5 cm * 8 cm holes, Section 2.3.2) ensured a possibility for tactile (snout-to-snout touches) and audial contacts, thought to be the main factors for establishing a social affiliation in rats, e.g., [39,40,41]. Audial communications had been recorded and remained to be published; here we measured social approaches only by the parameters of snout-to-box contacts.

A distinction between curiosity and social drive can be made assessing the contact time [34]. Short (less than 6 s) bouts of investigation reflect curiosity per se, while longer well-developed contacts relate to communicative behavior. Thus, our results show both a lack of curiosity and a deficiency in communicative drive in KM rats (Section 3.2).

The Novel Object Recognition (NOR) task is used to assess cognitive abilities, in particular recognition memory, in models of CNS disorders in rodents [47]. This test is based on the spontaneous tendency of rodents to spend more time studying a new object than a familiar one. The choice to study a new object reflects the use of learning and recognition memory. In our experiments, KM rats displayed decreased contacts of animated objects (i.e., boxes with a conspecific), but spent a longer time inspecting inanimate small objects and interacting with them (Figure 8(a2,b2)). It points to a social specificity of the observed investigatory deficits in KM rats.

In the social dominance test, KM rats demonstrated a poor behavioral repertoire, choosing a passive progression strategy. However, in this test, it is just the strategy that turned out to be more advantageous, since it did not allow the opponent pushing such a rat out. KM rats could move forward without pushing out the opponent, but rather took up a space after its active retreat. Wistar rats preferred to escape confrontations, and it seemed as they voluntarily left the pipe. Thus, KM rats did not show active dominance, but also did not allow dominating them. Similar results have been reported recently in the “social dominance in a tube test” for high-ranking Syngap+/Δ-GAP rats (heterozygous for the C2/GAP domain deletion, with ASD-like traits). The Syngap+/Δ-GAP individuals won significantly more competitions against high-rank wild type rats [64]. We can conclude that the consistent social deficits seen in KM rats (Section 3.2) were not attributable to a putative social defeat experience or submissive social traits.

An obvious limitation of the present work is the lack of female subjects in the study. Although ASD has a profound male prevalence in clinical practice [65], the female ASD also needs extensive attention of basic and clinical scientists. However, the official breeder of the KM rat strain, Biological Faculty of Moscow State University, restricts a dissemination of female KM individuals, for sake of the guaranteed genetic purity. So, experimentations in female KM rats, as well as early ontogenetic studies, are not feasible for external teams.

In the present experiments, we did not distinguish clearly between the sensory modalities engaged in social affiliation and its disruption. Visual inputs seem to be minimal since we used dark perforated containers with gap-like communication windows (Section 3.2.2 and Section 3.2.3). Presumably, the tactile contacts provided the core component of social reward in rats [39,66]. But it is also possible that reciprocity of tactile or audial contacts is of importance. To clarify this point, experimentation with toy rats or a deactivation of stimuli rats by a light narcosis would be useful.

A clear advantage of the study is the experimental proof that severe but still rare seizures do not cause an obligatory decay in sociability, which would be a strong concern for many caregivers of ASD children. Moving further and applying more seizure provocations, we may find the time point of a seizure-related regress in ability to develop social contacts. This would allow dissecting possible neurodegenerative mechanisms of subserving a social withdrawal; thus, this knowledge, accumulated in epileptology, might be of benefit for the understanding ASD-related neurological issues. For this purpose, we need various animal models in order to apply seizure provocations in a controlled way and monitor emotional and motivational changes, paralleling the time course of epilepsy.

In experimentations with laboratory animals, it is important to level out possible contextual artifacts by using a battery of behavioral tests [29]. Multidimensional ethological analysis, applied to behavioral components of social deficits in laboratory animals, and a concise translation of the knowledge to a broader context of human neuropathology would bridge a way to find effective treatments for ASD-related clinical issues.

The obtained results confirmed and expanded our previous conclusions about social deficiency in KM rats. One of the oldest inbred models of genetic epilepsy has translational potential for studying the neurophysiology and pharmacology of ASD-like social deficiencies. A special clinical domain of the autistic spectrum, characterized by hypolocomotion and excessive freezing [67], can be modeled in the KM strain. It is important to note that anxiety and locomotor deficits could be normalized by extra-long habituation and/or repeated testing; the same was not observed for social contacts deficiency in these rats. Thus, hypolocomotion was experimentally dissociated from contact deficiency. It means that we need more than just an environmental habituation to treat social contact deficits.

Importantly, severe but rare audiogenic convulsions, as well as sound provocations per se, did not aggravate contact deficits in the rats tested. This suggests that between the seizure onset and an impending seizure-related social regress, there is a time window to find an effective anti-epileptic therapy.

## Figures and Tables

**Figure 1 biomedicines-11-02566-f001:**
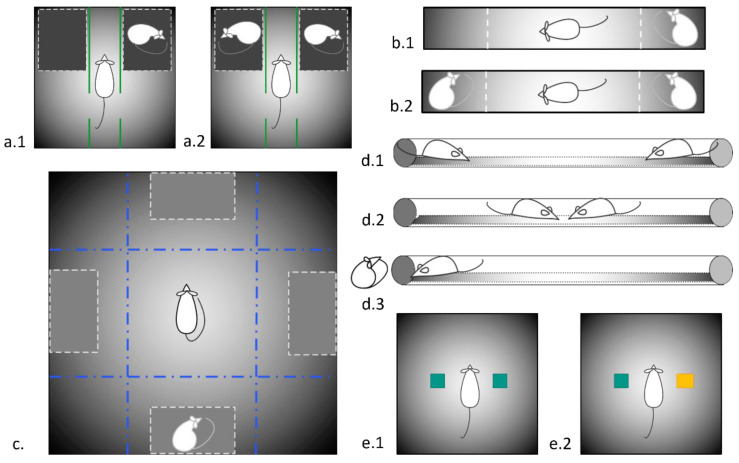
Schemes of experimental set-ups used in the study. (**a.1**) Three-chambered social preference test; (**a.2**) Three-chambered social novelty test; (**b.1**) The two-choice test for social preference/social novelty, 1st session, social preference; (**b.2**) The two-choice test for social preference/social novelty, 2nd session, social novelty (**c**) Socially enriched open field; (**d.1**–**d.3**) different stages of social dominance test; (**e.1**,**e.2**) 1st and 2nd session of two-object/novel object recognition test.

**Figure 2 biomedicines-11-02566-f002:**
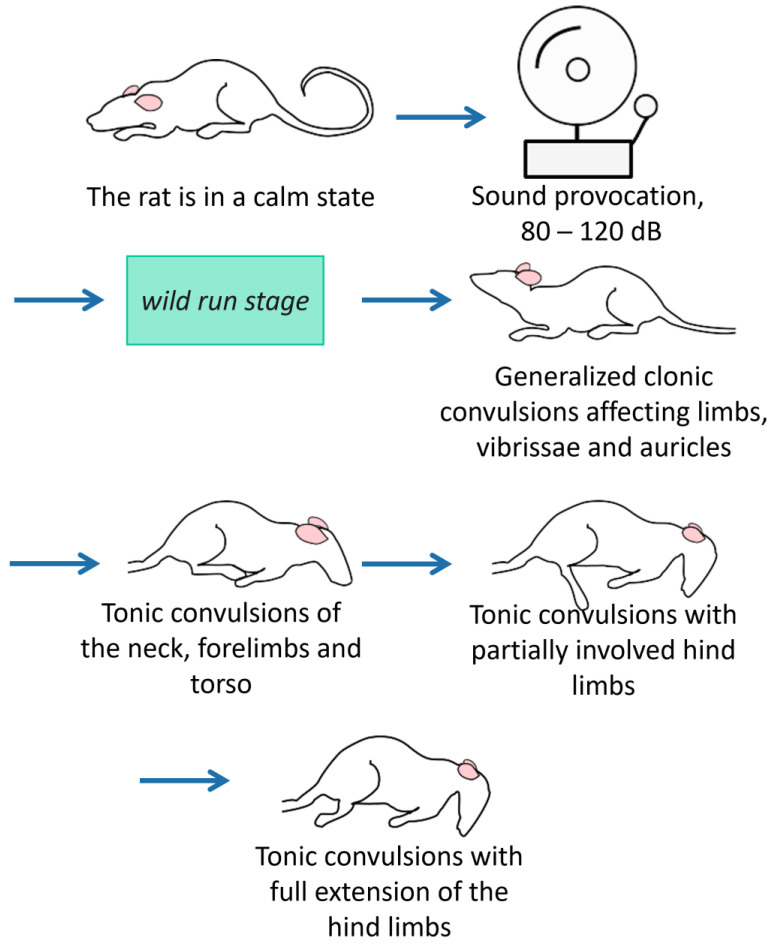
The stages of audiogenic fits in KM rats. The seizure fits start with motor excitation (wild running with a loss of visual control), quickly proceed to clonic convulsions on a side and on the belly, and end up with tonic phases (with the tonic limbs’ extension). These stages of audiogenic fits are: wild run, clonic seizures on a belly, tonic seizure. In total, 100% of the KM rats develop the maximal stage of audiogenic fits in response to the sound administration.

**Figure 3 biomedicines-11-02566-f003:**
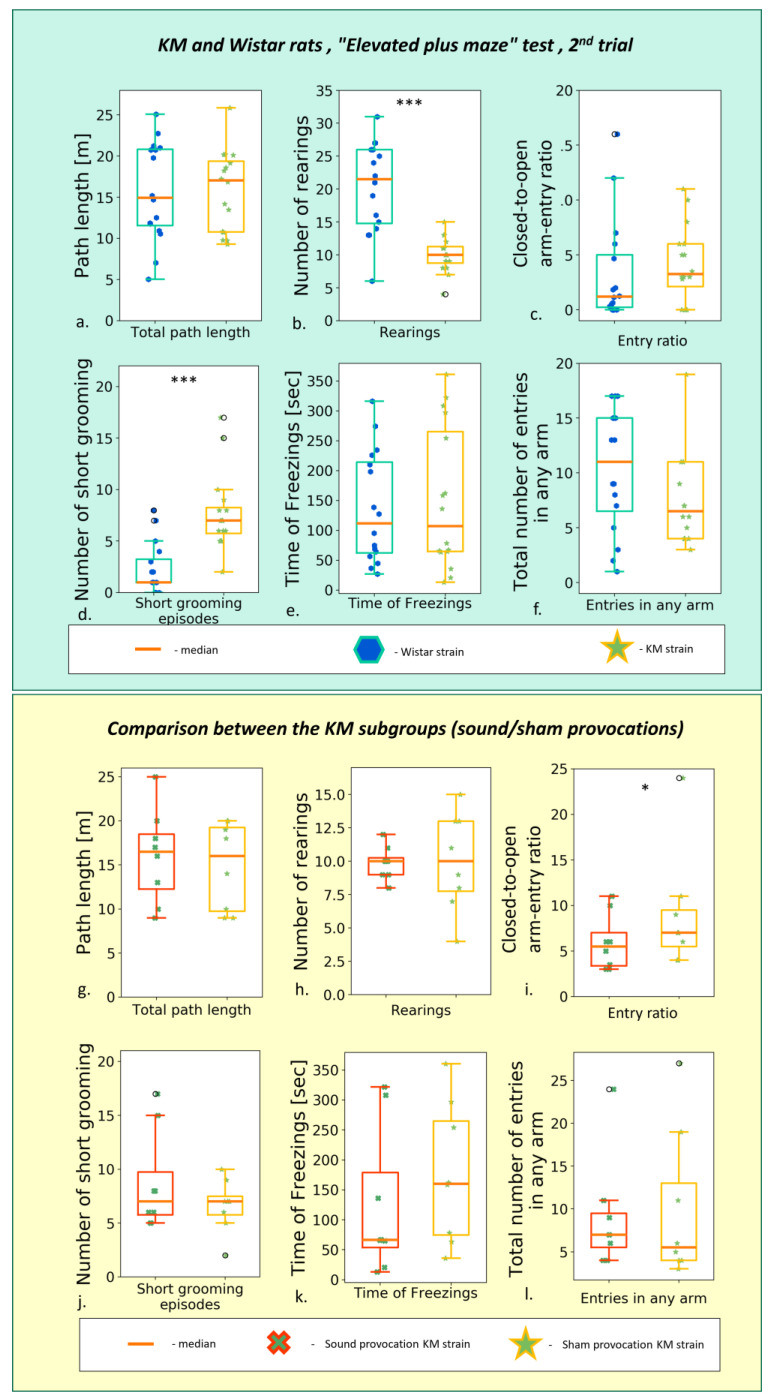
Behavior in the second (post-provocation) “elevated plus maze” test, comparison between the rat strains (upper panel, Wistar vs. KM) and KM subgroups (lower panel, sound provocation vs. sham provocation). Upper panel: (**a**) The total path travelled; (**b**) number of rearings t; (**c**) closed-to-open arm-entry ratio; (**d**) number of short (facial) grooming bouts; (**e**) total time of freezing behavior; (**f**) total number of entries in any arm. Lower panel: (**g**) The total path travelled; (**h**) number of rearings t; (**i**) closed-to-open arm-entry ratio; (**j**) number of short (facial) grooming bouts; (**k**) total time of freezing behavior; (**l**) total number of entries in any arm. Data are represented as the box plot and whiskers (indicating variability outside the upper and lower quartiles). Statistical significance is indicated as follows: *** *p* < 0.001, * *p* < 0.05.

**Figure 4 biomedicines-11-02566-f004:**
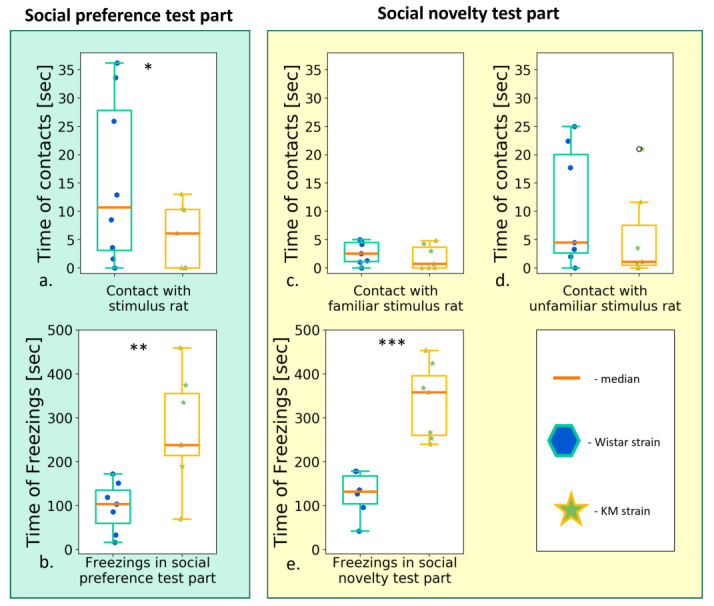
Contact activity of Wistar (blue points) and KM (green stars), in the test of “two choice social preference test”. (**a**) Time of contact with an unfamiliar stimulus rat; (**b**) total freezing time of rats during the test; (**c**) time of contact with a familiar stimulus rat during the “social novelty” part; (**d**) time of contact with an unfamiliar stimulus rat during the “social novelty” part; (**e**) total freezing time during the “social novelty” session. Data are represented as the box plot and whiskers (indicating variability outside the upper and lower quartiles). Statistical significance is indicated as follows: *** *p* < 0.001, ** *p* < 0.01, * *p* < 0.05.

**Figure 5 biomedicines-11-02566-f005:**
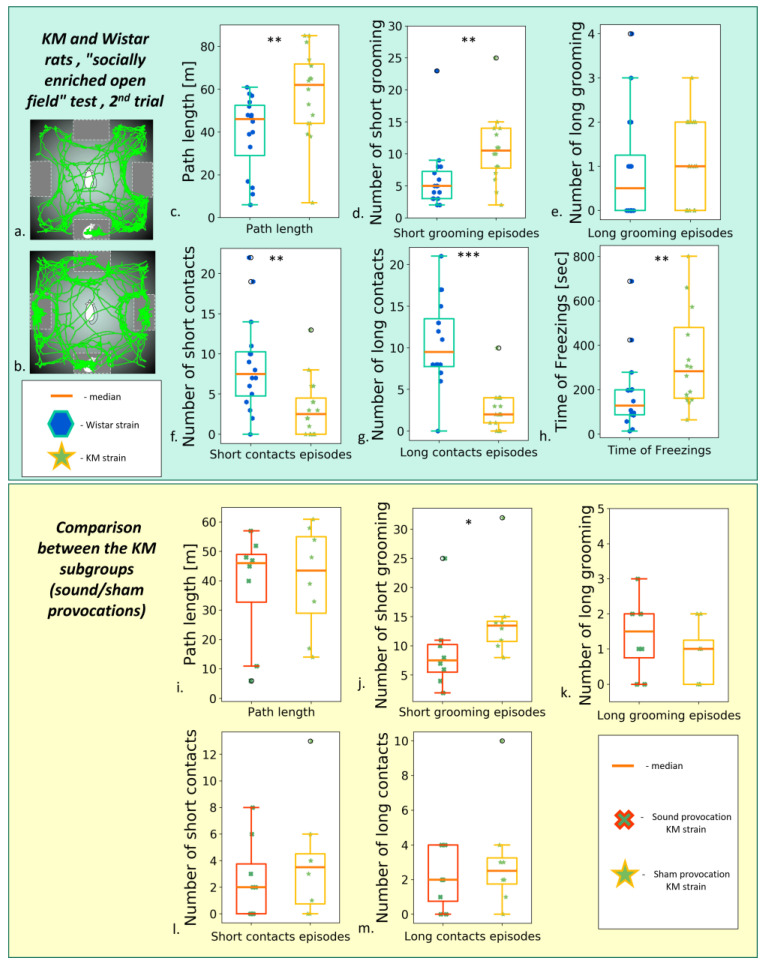
Behavior in the “socially enriched open field” test, the second test (post-provocation). (**a**) An example of Wistar rat trajectory; (**b**) an example of KM rat trajectory. Upper panel: comparisons between the strains. (**c**) The total path length; number of short (**d**) and long (**e**) grooming; the number of short (**f**) and long (**g**) of snout-to-box contacts of the freely moving experimental animals; (**h**) the total freezing time. Lower panel: comparison between seizure-experienced and seizure-naïve KM subgroups. (**i**) The total path length, comparison between the KM subgroups. Number of short (**j**) and long (**k**) grooming; number of short (**l**) and long (**m**) snout-to-box contact, comparison between the KMs subgroups. Data are represented as a box plot and whiskers (indicating variability outside the upper and lower quartiles). Statistical significance is indicated as follows: *** *p* < 0.001, ** *p* < 0.01, * *p* < 0.05.

**Figure 6 biomedicines-11-02566-f006:**
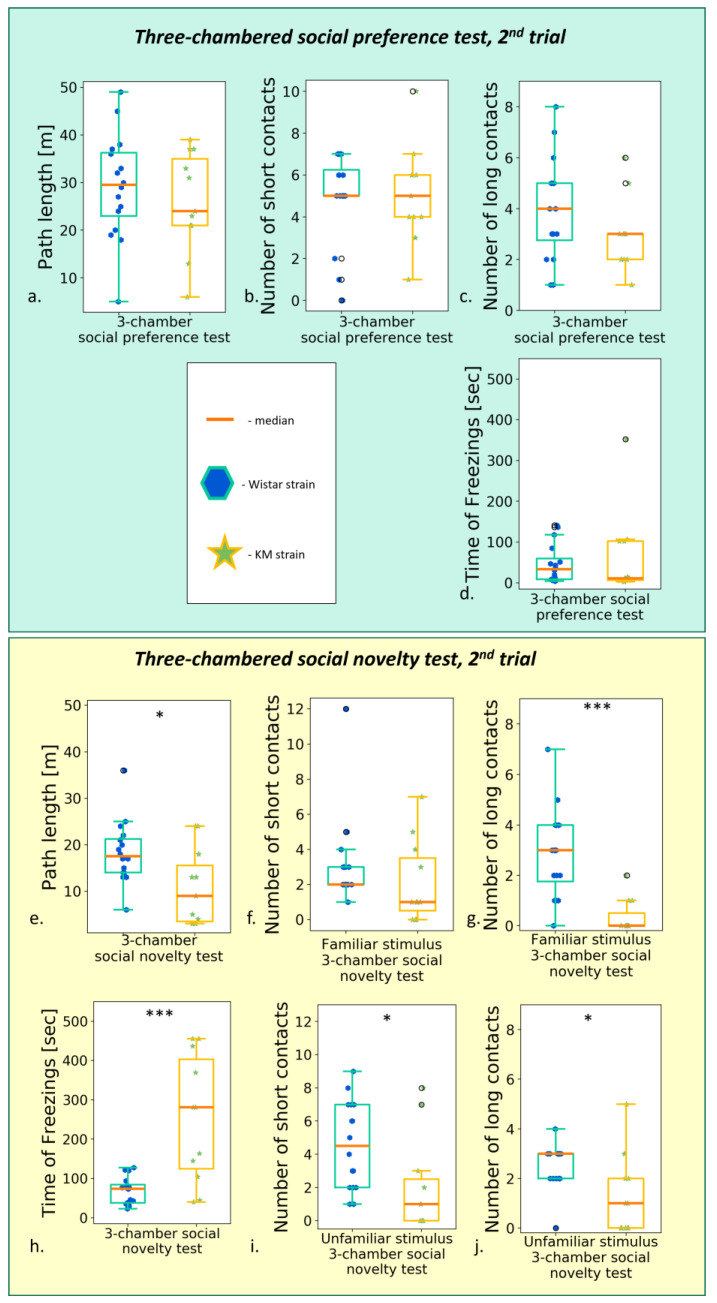
Behavior in the “social preference/social novelty” tests, post-provocation, between groups comparison (Wistar vs. KM rats). The “social preference” session: (**a**) total path travelled; (**b**) number of short contacts; (**c**) number of long contacts; (**d**) total time of freezing s. The “social novelty” session: (**e**) total path travelled; (**f**) number of short contacts with “old” stimulus; (**g**) number of long contacts with “old” stimulus; (**h**) time of freezing response; (**i**) number of short contacts with “new” stimulus; (**j**) number of long contacts with “new” stimulus. Data are represented as a box plot and whiskers (indicating variability outside the upper and lower quartiles). Statistical significance is indicated as follows: *** *p* < 0.001, * *p* ≤ 0.05.

**Figure 7 biomedicines-11-02566-f007:**
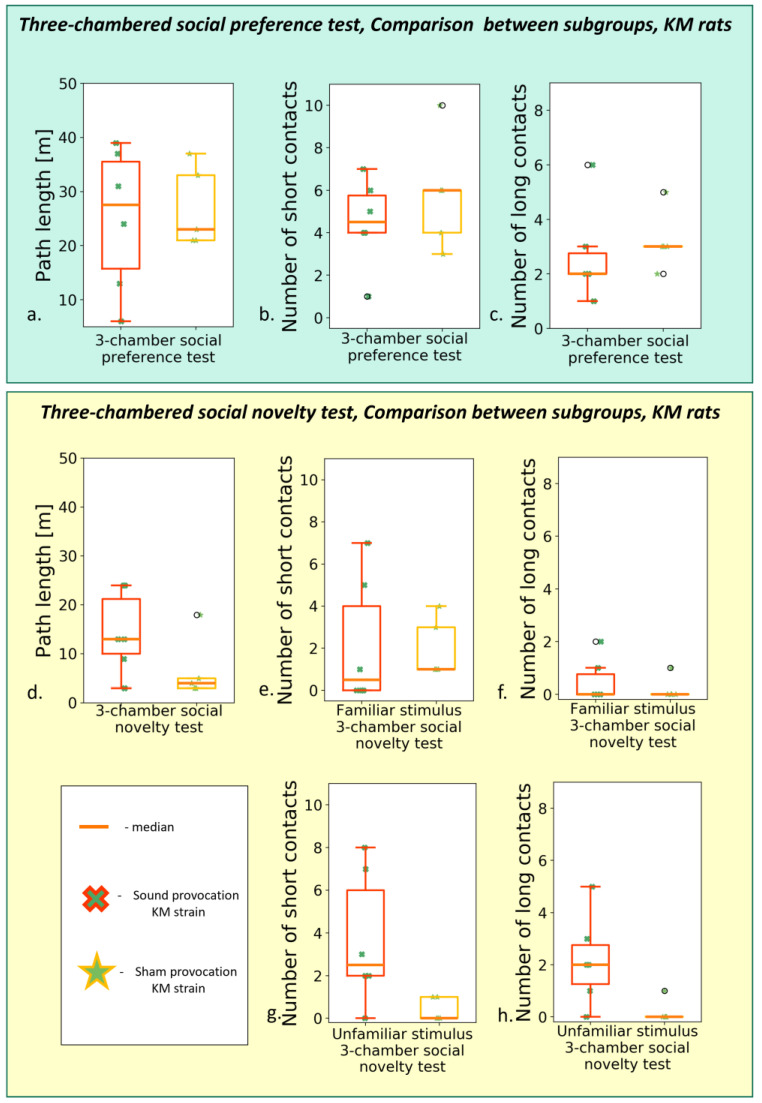
Behavior in the “social preference/social novelty” tests, post-provocation; comparison between the KM subgroups. The “social preference” session: (**a**) total path length; (**b**) number of short contacts; (**c**) number of long contacts. The “social novelty” session: (**d**) total path length, (**e**) number of short contacts with “familiar” stimulus; (**f**) number of long contacts with “familiar” stimulus; (**g**) number of short contacts with “unfamiliar” stimulus; (**h**) number of long contacts with “unfamiliar” stimulus. Data are represented as a box plot and whiskers (indicating variability outside the upper and lower quartiles). No significant difference was seen between the subgroups.

**Figure 8 biomedicines-11-02566-f008:**
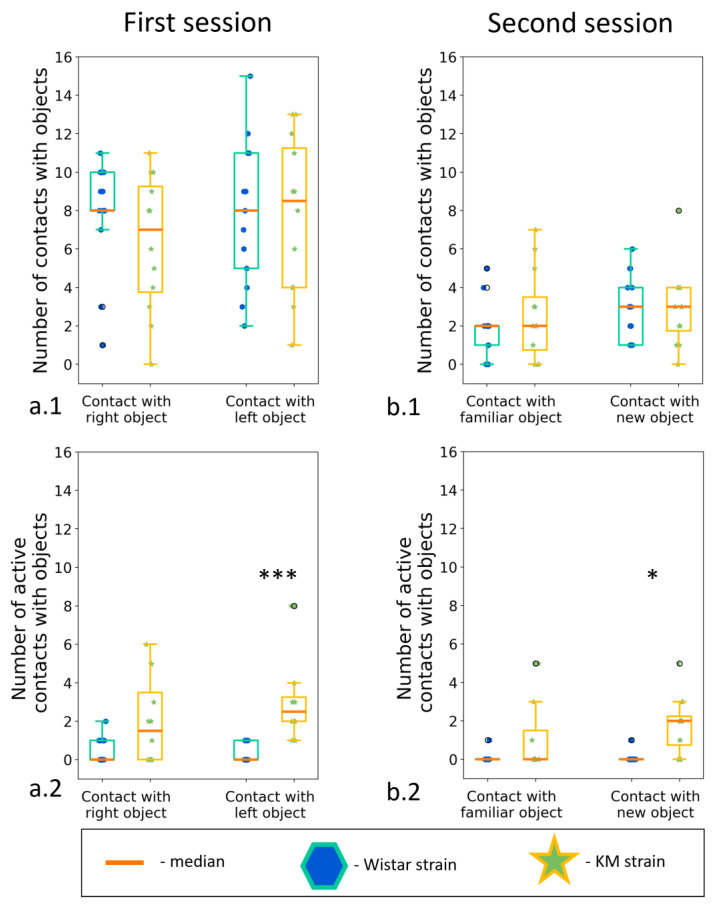
The “two objects novel object test”, between strain comparisons. Investigation of new and familiar objects. (**a.1**) Contacts with a new object; (**a.2**) active contacts with a new object; (**b.1**) contact with an “old” object; (**b.2**) Active contact with an “old” object. Data are represented as a box plot and whiskers (indicating variability outside the upper and lower quartiles). Statistical significance is indicated as follows: *** *p* < 0.001, * *p* < 0.05.

**Figure 9 biomedicines-11-02566-f009:**
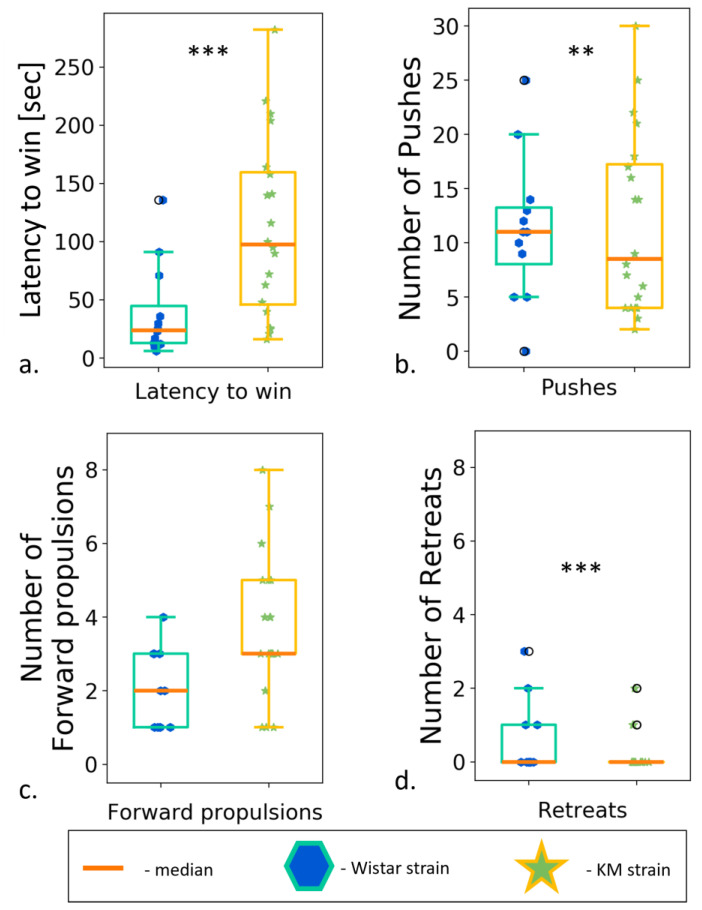
Social dominance test, between strain comparisons. (**a**) Latency to win; (**b**) number of pushes; (**c**) number of forward propulsions; (**d**) number of retreats. Data are represented as a box plot and whiskers (indicating variability outside the upper and lower quartiles). Statistical significance is indicated as follows: *** *p* < 0.001, ** *p* < 0.01.

## Data Availability

The experimental data are fully reported in the manuscript and Appendix A. Additional information is available upon a reasonable request.

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
