# Peer review of "Audiogenic Seizures and Social Deficits: No Aggravation Found in Krushinsky–Molodkina Rats"

_biomedicines, 2023, doi:10.3390/biomedicines11092566_

Round 1

Reviewer 1 Report

This work is well performed and the results are interesting.

I suggests some minor changes:

1) A correction for multiple comparisons should be added to the results section.

2) A brief comment on the strengths and limitations of the work should be added in the discussion.

3) Abbreviations such as ASD (autism spectrum disorder) and ADHD (attention deficit hyperactivity disorders) should be defined in the text when they are use for the first time

4) References citation should be corrected. For example [4, 8, 5, 6, 7] should be mentioned as [4-8]

Author Response

The Authors are grateful for the positive comments and useful comments. All the points raised have been encountered, namely:

- 1) A correction for multiple comparisons should be added to the results section.

         - We have applied Bonferroni corrections to the between-subgroups comparisons (sound-stressed and sham-stressed subgroups of KM rats; Tables 1D, 3C, 4C). The corrections were added to all related Supplementary Tables, and corresponded Figures. The corresponding text (Results and Discussions) has been changed. But since the specific behavioral traits (namely, the social deficits) was quite reproducible in different cohorts of KM rats, and in different experimentations, the probability of a false-positive result seems to be quite low

-2) A brief comment on the strengths and limitations of the work should be added in the discussion.

         - thanks for the important reminder. The following sentences were added:

“An obvious limitation of the present work is the lack of female subjects in the study. Although ASD has a profound male prevalence in clinical practice [43], the female ASD  also need an extensive attention of basic and clinical scientists. However, the official breeder of KM rat strain, Biological Faculty of Moscow State University, restricts a dissemination of female KM individuals, for sake of the guaranteed genetic purity. So, the experimentations in female KM rats, as well as early ontogenetic studies, are not possible for external teams.

A clear advantage of the study is the experimental proof that severe but still rare seizures do not cause an obligatory decay in sociability, which would be a strong concern for many parents of ASD children”

3) Abbreviations such as ASD (autism spectrum disorder) and ADHD (attention deficit hyperactivity disorders) should be defined in the text when they are use for the first time

- this has been corrected

4) References citation should be corrected. For example [4, 8, 5, 6, 7] should be mentioned as [4-8]

- references have been checked

Reviewer 2 Report

GENERAL COMMENTS

For this reviewer, the “three-chambered social preference test” (quotation marks by authors; fig 1a.1) is flawed as an experimental paradigm because it does not compare a live object with an inanimate one (i.e., dummy rat, toy rat) with the same or approximate sensory properties – size, color, preferably also on a bedding smelling of a rat). The experimental paradigm used by the authors compares a live caged rat only to an EMPTY cage as one of the three “chambers” within the text box.

Do the authors have data to prove that rats do prefer a live rat to in animate (toy) rat? This has been shown for rats and another rodent genus in another test for the attractiveness of stimuli associated with social interaction (see, e.g., DOI: 10.1097/FBP.0000000000000223  and doi: 10.3389/fnbeh.2011.00080) . If not, the authors must discuss this lack of proof as a major shortcoming of their experimental approach – or quote such proof if obtained by other groups.

line 13, throughout the manuscript, e.g., line 502:  Please use “inanimate” instead of “unanimated” throughout the manuscript.

In their discussion (line 416ff), the authors emphasize “evident locomotion” as a feature of KM rats. To this reviewer, “freezing” does NOT equal “hypolocomotion”. The authors must discuss this issue.

SPECIFIC ITEMS

Freezing: fig4 showss considerably increased freezing time in social preference test (fig 4b) and in social noevelty test (fig 4e) in KM rats and a smaller but still stastically significant effect in the elevated plus maze (fig 3e). Also, the authors should use the SAME measure, i.e., either freezing TIME or freezing EPISODES in all tests (fig 3e, fig 4b and fig 4e).

fig5: Why was freezing not documented in the “socially enriched open field test” (quotation marks by authors)? Because there wasn’t any – which would be discrepant to the rats’ behavior in the other tests shown in fig4?

fig4 vs (figs 6 and 7): Why did the authors differentiate between “short” and “long” contacts in the 3-chambered social preference test (figs 6 and 7) and not in the 2-choice social preference test? Please justify or correct.

moderate editing recommended

Author Response

The authors are grateful for the extensive comments and discussion of the results presented. We do hope that the manuscript has been improved.

  • “For this reviewer, the “three-chambered social preference test” (quotation marks by authors; fig 1a.1) is flawed as an experimental paradigm because it does not compare a live object with an inanimate one (i.e., dummy rat, toy rat) with the same or approximate sensory properties – size, color, preferably also on a bedding smelling of a rat). The experimental paradigm used by the authors compares a live caged rat only to an EMPTY cage as one of the three “chambers” within the text box.”

  • we do agree, what EMPTY cages might be distractive for rats; that’s why all the cages used, even “empty” ones, contained bedding material from the homecage of the stimulus rats, to unify the olfactory cues. It was described in the method section (lines 135, 153). We do not use toy rats, since the toys might be smelling brightly for the rodents, but not for us – so, it would be one more factor to complicate the design. In our set-ups, the animals were able to contact with snouts, sniff, and vocalize – not to groom each other or bite. So, tactile modalities were set to be minimal.

  • Do the authors have data to prove that rats do prefer a live rat to in animate (toy) rat? This has been shown for rats and another rodent genus in another test for the attractiveness of stimuli associated with social interaction (see, e.g., DOI:10.1097/FBP.0000000000000223 and doi:10.3389/fnbeh.2011.00080). If not, the authors must discuss this lack of proof as a major shortcoming of their experimental approach – or quote such proof if obtained by other groups.

  • yes, there are studies of the same design reported in the literature. Although there is still no commonly accepted “golden standard” for a design of sociability tests, variations of “socially enriched open fields” and “social preference/social novelty” tests are reported are reported (f.e.: DOI: 10.1098/rsob.200306; https://doi.org/10.1016/j.yebeh.2015.08.039; doi.org/10.1038/s41467-020-19569-0). The authors used triangle mesh cages in the corners, or wire cylinders in the middle of the arena. We decided to use the same containers, as used for regular transportations of individual rats in the lab, to reduce all possible novelty factors, not related to social stimuli. Four boxes in the “socially enriched open field” test were used to counterbalance possible uncontrollable environmental factors (like smell or ultrasound flows). The “populated” box was set in the middle of each wall with the same probability, so – the observed behavior should be free of unwanted artifacts. One of the practical purposes for our team was to compare available setups, and choose the most robust one. In our hands, it appears to be “the social novelty test”.

We added the following sentence to the text, for a better introduction of the methodological approach:

“The main idea of the social tests was to allow a freely moving test animal to explore the test arena with an encaged unfamiliar conspecific. As it was shown previously, normal rats demonstrate a clear preference to stay in a neighbourhood of a sheltered new rat, rather than nearby empty cages [15-17].”

  • line 13, throughout the manuscript, e.g., line 502: Please use “inanimate” instead of “unanimated” throughout the manuscript.
  • this was corrected, thank You.

-“In their discussion (line 416ff), the authors emphasize “evident locomotion” as a feature of KM rats. To this reviewer, “freezing” does NOT equal “hypolocomotion”. The authors must discuss this issue.”

  • hypolocomotion is not equal to pronounced freezing, indeed. The hypolocomotor trait in KM rats is quite known (https://doi.org/10.46867/C47K5F), and it complicates the interpretation of the results. That’s why we reject the two choice social preference/social novelty test – the animals are too stressed and demonstrate enormous amount of freezing bouts. Also, we report percentages of path travelled/time spent around the stimuli boxes (Tables 3A, 3C, 3D) in the open field. Finally, in the present work we have shown, that hypolocomotion in KM rats might be normalized, by an extra-long habituation. Many of our 2nd trials (post-provocation) demonstrated, that in a month of weekly experimentations, KM rats started to move almost the same active, as normal Wistars did after single short habituation sessions.

SPECIFIC ITEMS

  • Freezing: fig4 shows considerably increased freezing time in social preference test (fig 4b) and in social noevelty test (fig 4e) in KM rats and a smaller but still stastically significant effect in the elevated plus maze (fig 3e). Also, the authors should use the SAME measure, i.e., either freezing TIME or freezing EPISODES in all tests (fig 3e, fig 4b and fig 4e).

  • the inconsistency was corrected; now all the Figures and Tables contain the same measures

fig5: Why was freezing not documented in the “socially enriched open field test” (quotation marks by authors)? Because there wasn’t any – which would be discrepant to the rats’ behavior in the other tests shown in fig4

  • we included the measure as well; the freezing chart is on its place on Fig.5h (for the 1st trial see supplementary Table 3A).

  • fig4 vs (figs 6 and 7): Why did the authors differentiate between “short” and “long” contacts in the 3-chambered social preference test (figs 6 and 7) and not in the 2-choice social preference test? Please justify or correct.
  • the test design appears to be too stressful for rats, they almost did not develop prolonged contacts (supplementary Table 2). We included the included a corresponding sentence to the Subsection 2.3.1:

“In this paradigm, the rats mainly demonstrated short (<6sec) contacts, presumably reflecting general curiosity but not a social drive [16]. The prolonged, developed (> 6sec) social investigations were almost absent in this condition (mean numbers of prolonged contacts = 0±0, and 0.9±0.5 for KM and Wistar groups, respectively; see supplementary Table2). Short contacts summed up to a lower time in KM rats (p<0.05, Fig.4a; supplementary Table. 2)”

Round 2

Reviewer 2 Report

The authors have addressed a number of the reviewer’s concern - alas, more eloquently in their response to the reviewer than in the manuscript itself. However, the folloying issues have not been addressed by the authors yet. Therefore, in oder to be acceptable for publication, the following remaining issues MUST be discussed IN THE MANUSCRIPT   1) Do the authors have data to prove that rats do prefer a live rat to in animate (toy) rat? This has been shown for rats and another rodent genus in another test for the attractiveness of stimuli associated with social interaction (see, e.g., DOI:10.1097/FBP.0000000000000223 and doi:10.3389/fnbeh.2011.00080). If not, the authors must discuss this lack of proof as a major shortcoming of their experimental approach – or quote such proof if obtained by other groups.

2) In their discussion (line 416ff), the authors emphasize “evident locomotion” as a feature of KM rats. To this reviewer, “freezing” does NOT equal “hypolocomotion”. The authors must discuss this issue.

very minor improvements could still be made

Author Response

The authors are grateful for the discussion of interesting scientific and methodological aspects. We tried to extend and clarify the points raised.

(The comment: The authors have addressed a number of the reviewer’s concern - alas, more eloquently in their response to the reviewer than in the manuscript itself. However, the folloying issues have not been addressed by the authors yet. Therefore, in oder to be acceptable for publication, the following remaining issues MUST be discussed IN THE MANUSCRIPT   1) Do the authors have data to prove that rats do prefer a live rat to in animate (toy) rat? This has been shown for rats and another rodent genus in another test for the attractiveness of stimuli associated with social interaction (see, e.g., DOI:10.1097/FBP.0000000000000223 and doi:10.3389/fnbeh.2011.00080). If not, the authors must discuss this lack of proof as a major shortcoming of their experimental approach – or quote such proof if obtained by other groups.)

  • lines 494-508: The contact behavior of test rats depended on a space available for interaction, as well as on ability to touch and smell the guest rats [(doi: 10.3389/fnbeh.2011.00080).] A close spatial confinement of interacting rats led to a decreased contact motivation; while their partition by barred walls preserved it. Tactile interactions were essential for social reward, whereas visual contacting via glass screen was not [(doi: 10.3389/fnbeh.2011.00080)]. In our experiments, perforated walls allowed the olfactory inspections of all the containers, which were unified in smelling by bedding compound from the guest rats’ homecages. So the only presumed difference between the cages was a presence of an encaged conspecific. The communication windows (1.5cm*8cm holes, Subchapter 2.3.2) ensured a possibility for tactile (snout-to-snout touches) and audial contacts, thought to be the main factors for establishing a social affiliation in rats [f.e., doi:10.1371/ journal.pone.0001365; doi: 10.3389/fnbeh.2011.00080; doi.org/10.1016/j.cobeha.2022.101129]. Audial communications had been recorded and remained to be published; here we measured social approaches only by the parameters of snout-to-box contacts.

and futher,

  • lines 549-556: In the present experiments, we did not distinguish clearly between the sensory modalities engaged in social affiliation and its disruption. Visual inputs seem to be minimal, since we used dark perforated containers with gap-like communication windows (Subchapters 3.2.2 and 3.2.3). Presumably, the tactile contacts provided the core component of social reward in rats [doi: 10.1111/j.1369-1600.2010.00285.x; doi: 10.3389/fnbeh.2011.00080]. But it is also possible, that reciprocity of tactile or audial contacts is of importance. To clarify this point, experimentation with toy rats or a rat under narcosis would be useful.

(the comment: In their discussion (line 416ff), the authors emphasize “evident locomotion” as a feature of KM rats. To this reviewer, “freezing” does NOT equal “hypolocomotion”. The authors must discuss this issue.)

  • lines 438-441: “To overcome potential artifacts, caused by the excessive freezing response and hypolocomotion, we used the percentages of time spent near the stimulus’ box, and the percentages of path travelled around it (supplementary Tables 3A and 3C).”

and also,

  • lines 453-455: “Thus, the hypolocomotion and social deficits in KM rats were dissociated in the present experiments. It is possible, that longer repetitive social exposures would normalize contact with conspecifics, in its turn.”

Round 3

Reviewer 2 Report

Nice job. Accept in present form.

Author Response

We are thankful for Your attentive review and believe that the manuscript has been improved.